# Advances in Liquid Biopsy Technology and Implications for Pancreatic Cancer

**DOI:** 10.3390/ijms24044238

**Published:** 2023-02-20

**Authors:** Alexander G. Raufi, Michael S. May, Matthew J. Hadfield, Attila A. Seyhan, Wafik S. El-Deiry

**Affiliations:** 1Division of Hematology/Oncology, Department of Medicine, Lifespan Health System, Providence, RI 02903, USA; 2Legorreta Cancer Center, Brown University, Providence, RI 02903, USA; 3Joint Program in Cancer Biology, Brown University, Providence, RI 02903, USA; 4Division of Hematology/Oncology, Columbia University Irving Medical Center, New York, NY 10032, USA; 5Department of Pathology and Laboratory Medicine, Warren Alpert Medical School, Brown University, Providence, RI 02903, USA

**Keywords:** liquid biopsy, pancreatic ductal adenocarcinoma, ctDNAs, exosomes, miRNAs, CTCs

## Abstract

Pancreatic cancer is a highly aggressive malignancy with a climbing incidence. The majority of cases are detected late, with incurable locally advanced or metastatic disease. Even in individuals who undergo resection, recurrence is unfortunately very common. There is no universally accepted screening modality for the general population and diagnosis, evaluation of treatment response, and detection of recurrence relies primarily on the use of imaging. Identification of minimally invasive techniques to help diagnose, prognosticate, predict response or resistance to therapy, and detect recurrence are desperately needed. Liquid biopsies represent an emerging group of technologies which allow for non-invasive serial sampling of tumor material. Although not yet approved for routine use in pancreatic cancer, the increasing sensitivity and specificity of contemporary liquid biopsy platforms will likely change clinical practice in the near future. In this review, we discuss the recent technological advances in liquid biopsy, focusing on circulating tumor DNA, exosomes, microRNAs, and circulating tumor cells.

## 1. Introduction

Pancreatic ductal adenocarcinoma (PDAC) is a highly aggressive malignancy characterized by an unparalleled mortality-to-incidence ratio. In 2023, it is estimated that 64,050 individuals will be diagnosed and 50,550 individuals will die from PDAC in the United States [1]. The five-year survival rate has only recently increased to double-digits (~11%), but with a rising incidence and limited effective treatment options, PDAC is projected to become the second-leading cause of cancer-related death by 2030 [1].

Surgery represents the only curative treatment modality for PDAC and therefore early detection has the potential to significantly improve outcomes. Unfortunately, however, due to the insidious onset and lack of effective screening tools the vast majority of cases (82%) are detected late, with unresectable locally advanced or metastatic disease. Furthermore, even when a complete resection is performed, recurrence occurs in approximately 80% of cases, at which time the disease is almost universally fatal [2,3,4]. Adjuvant chemotherapy reduces the risk of recurrence and modified FOLFIRINOX (5-fluorouracil, leucovorin, irinotecan, and oxaliplatin) is the standard of care adjuvant regimen, demonstrating an overall survival benefit of 19.4 months over gemcitabine alone (54.4 months vs. 35.0 months) [3].

In the metastatic setting, palliative systemic chemotherapy, which can mitigate symptoms while simultaneously prolonging life, plays a dominant role. There are currently two standard of care first-line treatment options, FOLFIRINOX, as mentioned above, and the combination of gemcitabine and nab-paclitaxel. Compared to single-agent gemcitabine, FOLFIRINOX improves the median overall survival by 4.3 months (11.1 months vs. 6.8 months) [3,5] whereas the addition of nab-paclitaxel to gemcitabine improves median overall survival by 1.8 months (8.5 months vs. 6.7 months) [6].

In addition to desperately needed new treatments, improved diagnostic tools to identify disease at earlier stages, guide therapy selection, determine treatment response, and predict recurrence are essential. Aside from high-risk individuals, universal screening of the general population is ineffective [7]. Currently, there is evidence supporting screening only for those with predisposing genetic conditions with a lifetime PDAC risk of >5–15%, high-risk pancreatic lesions, and a strong family history, typically with magnetic resonance imaging (MRI) or endoscopic ultrasound (EUS) [7]. The optimal timing and combination of these tests remains unknown. Methods for determining treatment response and recurrence are also limited to imaging and serum tumor markers, tests which often lack adequate sensitivity and specificity [8,9].

Liquid biopsy is a term used to describe several emerging technologies focused on using bodily fluids as sources of tumor-derived genetic material, other biological material such as proteins and metabolites, or tumor cells, for diagnostic, predictive, and prognostic purposes. Compared with traditional tissue biopsy, liquid biopsy provides several significant advantages (Figure 1) and is increasingly being used as a complementary tool in those who have previously undergone biopsy. Although peripheral blood is traditionally used for non-invasive sampling, urine, saliva, stool, ascitic fluid, pancreatic exocrine secretions, and portal venous blood are all potential sources of this material. Liquid biopsy components can be broken down into cell-free DNA (cfDNA), exosomes, circulating tumor cells (CTCs), microRNA (miRNA), and cell-free RNA (cfRNA) (Figure 2). Advances in DNA/RNA amplification and sequencing techniques, as well as capture of circulating tumor cells, exomes, and microvesicles have dramatically increased in recent years, making it soon possible integrate these technologies into clinical practice [10,11,12]. In this review, we will discuss the advances in methodologies for liquid biopsy in PDAC and potential implications for patient care.

## 2. ctDNA

The term cell free DNA (cfDNA) is used to describe extracellular DNA isolated from blood or other bodily fluids. cfDNA arising from malignant cells is more specifically referred to as circulating tumor DNA (ctDNA) and contains cancer-specific genetic alterations. cfDNA and ctDNA are released from normal and malignant cells either through cell death processes, such as apoptosis or necrosis, or through excretion [13]. Both can be isolated from plasma or serum, but plasma generally yields better sample quality due to decreased contamination from leukocyte DNA [14]. While cfDNA fragments are on average 166 base pairs (bps) in length, ctDNA is generally more fragmented, with an average length of about 140 bps [15] The relative amount of cfDNA that is ctDNA, referred to as the variant allele frequency (VAF), and can vary greatly. For example, in early stages of cancer the VAF is often less than 1%, whereas in the metastatic setting this is often much higher, with reports ranging from 5–80% depending on extent and location of disease [16,17]. Notably, many studies including patients with metastatic disease will report cfDNA levels as a surrogate for ctDNA when prognosticating or evaluating response to treatment [18].

Genomic profiling has revealed a high frequencies of a limited number of mutations occurring in Kirsten rat sarcoma virus (*KRAS*), tumor protein 53 (*TP53*), cyclin-dependent kinase inhibitor 2A (*CDKN2A*), and *SMAD4*. This makes PDAC an ideal disease to use ctDNA for screening, monitoring for treatment response or recurrence, and to guiding tumor-specific treatment [19,20]. The vast majority of PDAC tumors carry a mutation in or amplification of *KRAS* or inactivation of *TP53* (90% and 73% of cases, respectively) [20]. The next two commonly mutated genes are *CDKN2A* and *SMAD4* (35% and 31%, respectively) [20]. In addition, 10–20% of patients with PDAC will carry a germline mutation in a gene encoding a DNA damage response protein such as *ATM*, *BRCA1/2*, or *PALB2* [20]. Lastly, tumor mutational burden (TMB) and microsatellite instability (MSI) can also be determined using ctDNA, thus serving as a predictive tool for therapies targeting immune checkpoints [21,22]. Until recently, ctDNA sequencing methods lacked the required sensitivity and specificity for use in PDAC, however, this is beginning to change with recent technological advancements.

### 2.1. Methods for Detecting and Analyzing ctDNA

Techniques for ctDNA detection vary widely in regards to the types of genetic anomalies that can be identified, the VAF required, and cost. Most established methods use gene amplification to overcome the paucity of ctDNA in patients with PDAC, particularly in early-stage disease. Examples include real-time quantitative polymerase chain reaction (qPCR), digital PCR (dPCR), droplet digital PCR (ddPCR), and next-generation sequencing (NGS). With the exception of NGS, all of these techniques are limited by requiring predefined gene mutations of interest, which are amplified using prespecified sets of primers.

Unlike conventional PCR, qPCR monitors DNA amplification in real-time, with improved speed, reproducibility, and quantitation. However, qPCR is limited by low sensitivity and generally requires a VAF of 10% to successfully identify tumor-derived gene mutations. dPCR overcomes some of these limitations by separating a sample of DNA into thousands of compartments with zero, one, or multiple DNA strands [23]. Compartments are then amplified using parallel PCR reactions and from this the number of initial DNA strands is determined. Focusing on compartments containing a single parent DNA strand reduces the background noise associated with traditional PCR methods and enables the detection of tumor DNA at VAFs as low as 0.1% [24]. ddPCR is a method of dPCR which uses water-oil emulsion droplets to further fractionate a DNA sample into tens of thousands of droplets. PCR amplification is then performed independently in each droplet which further decreases background noise and allows for the detection of tumor DNA at VAFs as low as 0.01% [25].

Although more costly, NGS platforms have several advantages, including the ability to screen for unknown mutations, as well as structural and copy-number variations, which cannot be detected by PCR-based methods [26]. High-throughput analysis and whole-genome sequencing are also possible. Newer NGS technologies may even permit detection of malignant gene mutations at similar VAFs as those detectable with ddPCR [27,28,29]. One notable drawback of NGS is that it is currently more expensive than the aforementioned methods, typically costing thousands of dollars per sample. This cost, however, is dramatically decreasing. In addition, the limited mutational load in PDAC may reduce the need of NGS platforms [19]. NGS-based RNA sequencing of both tumor and peripheral blood using whole transcriptome sequencing platforms have also become commercially available, allowing for the identification of differentially expressed genes as well as identification of fusions, variant transcripts, and point mutations.

While dPCR, ddPCR, and NGS improve upon the many limitations of conventional PCR, none is able to detect epigenetic changes. Some of these changes, for example, methylation of CpGs clusters in promotor regions of tumor suppressor genes, have been implicated in tumorigenesis but are undetectable [30]. Recent advances in high-throughput quantitative methylation assays can now provide rapid and accurate identification of tumor DNA methylation using peripheral blood samples [31]. Furthermore, DNA methylation profiling has demonstrated reliability in predicting tumor of origin in patients with cancer of unknown primary [32]. More recently, epigenome and ATAC-sequencing have been used to simultaneously profile gene expression and open chromatin regions, and genome-scale DNA methylation (using reduced representation bisulfite sequencing; RBBS) [33,34]. In addition, isolating cell-free methylated-DNA using immunoprecipitation can be performed and coupled with NGS and PCR-based sequencing techniques, to improve specificity and reduce background noise [35]. In pancreatic cancer, differential hydroxymethylation of genes related to pancreas development or function (*GATA4*, *GATA6*, *PROX1*, *ONECUT1*, *MEIS2*), and cancer pathogenesis (*YAP1*, *TEAD1*, *PROX1*, *IGF1*) have also been shown to reliably identify pancreatic cancer from peripheral blood samples. As with DNA sequencing methods, the sensitivity and specificity of this method improves with more advanced cancer [36].

Lastly, several commercial liquid biopsy platforms capable of detecting ctDNA are now being used to guide clinical decisions for individuals with solid tumors. Examples include Guardant^TM^ (breast, colon, and lung cancers and multi-cancer detection) [37], FoundationOne^®^ (multi-cancer detection) [38], Signatera^TM^ (colorectal cancer) [39], Galleri^®^ (multi-cancer detection) [40], CancerSEEK (multi-cancer detection) [41] and Tempus^TM^ (multi-cancer detection) [42]. Additionally, Caris^®^ now provides bioinformatics testing of both circulating DNA and RNA [43].

### 2.2. ctDNA as a Screening Tool

Since surgery represents the only modality through which PDAC can be cured, detection of early-stage disease is paramount. Currently, no cost-effective screening tool exists for the general population; however, ctDNA detection platforms, which can be used longitudinally with frequent sampling requiring relative low blood volumes, may soon fill this void. Multiple studies using various platforms have investigated ctDNA as a potential screening tool (Table 1). A 2017 study compared ctDNA (quantified with ddPCR) to CA 19-9 and endoscopic ultrasound (EUS)/biopsy in 52 patients with PDAC, 10 patients with benign pancreatic tumors, and 6 patients with non-PDAC pancreatic malignancies. The investigators found that ctDNA had a sensitivity and specificity for PDAC of 65% and 75%, compared to 79% and 93%, for CA 19-9, and 73% and 88% for EUS/biopsy, respectively [44]. The relatively low sensitivity and specificity of ctDNA for detection of PDAC was thought to be due to a low VAF of ctDNA. Other groups have also reported correlations between *KRAS* VAF strongly and PDAC clinical stage [45]. This may be in part due to decreased numbers of cells undergoing apoptosis and necrosis in early-stage disease. Further complicating matters is the fact that ctDNA is rapidly cleared from the circulation by both endo- and exonuclease action and urinary excretion [46]. In fact the half-life of ctDNA ranges from as low as several minutes to two hours [47].

Combining ctDNA with established PDAC biomarkers as “composite or combination biomarkers,” may help to overcome these limitations. For example, one study defining positivity as having two of three of the following biomarkers: ctDNA, CA19-9, and CTCs, reported a sensitivity and specificity of 78% and 91%, respectively [44]. Another study found that combining ctDNA detection with optimized cutoffs of four tumor markers (CA19-9, CEA, hepatocyte growth factor (HGF), and osteopontin (OPN)) increased sensitivity for PDAC detection sensitivity from 30% to 64% with 99.5% specificity [57].

### 2.3. cctDNA to Guide Treatment and Monitor for Recurrence

Using ctDNA to guide treatment, predict, and detect tumor recurrence is of great interest and the number of studies incorporating ctDNA in these settings have dramatically increased (Table 1). Like somatic and germline sequencing, ctDNA can also be used to identify potentially actionable mutations. For example, ctDNA can be used to detect mutations in DNA damage response genes that predict benefit to PARP inhibitors or platinum chemotherapy. Using ctDNA, one can also identify fusions in *NTRK* and potentially actionable mutations in *HER2*, *AKT1*, *AKT2* and *CDK4*, facilitating prompt referral to relevant clinical trials [50]. Importantly, there is a high concordance in mutations detected in ctDNA and those found within primary tumors (66/66 in one study) [57].

As in resected colon cancer, detection of ctDNA following surgery has been shown to predict worse outcomes [60,61]. For example, a study using ddPCR demonstrated that persistence of ctDNA postoperatively predicted a median DFS and median OS of 8 months and 17 months, respectively, compared with 19 months and >30 months, respectively, in patients with undetectable ctDNA postoperatively [13,61,70]. Persistence of ctDNA following surgery is likely due to either residual local disease or occult micrometastatic disease and may support the use of additional chemoradiotherapy or chemotherapy. Several studies have also shown that ctDNA predicts a shorter disease-free survival (DFS) when detected prior to surgical resection of localized tumors [44,56,60,71]. Similarly, detection of ctDNA following completion of neoadjuvant chemotherapy has also been shown to predict recurrence following surgery [70].

Currently, monitoring for disease progression or recurrence is limited to imaging and tumor markers (e.g., CA19-9), both of which have limitations. Even for the most experienced radiologists, distinguishing between local recurrence and post-surgical or treatment-related inflammatory changes can be exceedingly difficult. Additionally, tumor markers are not expressed in many cases and lack specificity, occasionally increasing due to inflammation or radiation. As ctDNA technology continues to improve, it may soon be incorporated into routine use and eventually replace the use of imaging and tumor marker surveillance. Strikingly, a study from 2015 demonstrated that ctDNA could predict recurrence 6.5 months in advance of computed tomography [50]. Recent work by Sugimori et al., showed that fluctuations in *KRAS* VAF in patients with advanced pancreatic cancer undergoing treatment consistently correlated with increased risk of tumor progression and survival [58]. In this study of locally advanced and metastatic PDAC, of the 13 patients with detectable ctDNA at baseline treated with chemotherapy, 9 had disappearance of detectable ctDNA with treatment and all were found to have detectable ctDNA prior to or near the time of tumor recurrence by imaging [58]. Among PDAC patients with liver metastases, ctDNA trends have been successfully used to predict partial response, stable disease and disease progression, with ctDNA levels correlating with the number and size of metastases [58,63]. Combining ctDNA and with tumor markers may further increase these prognostic stratification [44,57,62]. In a 2020 study of 61 patients with metastatic PDAC, the use of the combination of CA19-9 with VAF, cfDNA concentration and cfDNA fragmentation improved prognostication of PDAC patients into high, medium and low risk groups for recurrence and death [62]. Improvements in these ctDNA technology may also reduce treatment-related morbidity, as early recognition of treatment resistance could spare patients unnecessary toxicity and facilitate more rapid changes in treatment plans.

## 3. Exosomes

Exosomes are membrane-derived extracellular vesicles, approximately 30 to 150 nanometers in size, containing a diverse variety of proteins, lipids, and nucleic acids [72]. Their content is reflective of the parent cells from which they originate. In the case of tumor cells, they are produced continuously. Recently, exosomes have demonstrated the ability to modulate the tumor immune microenvironment and their role in tumorigenesis is an area of active interest [73]. In fact, both pancreatic cancer cells and stromal cells have been shown to release and absorb exosomes. Nigri et al., demonstrated that through CD9, carcinoma-associated fibroblast-derived exosomes could induce cell migration and EMT in PDAC cells, subsequently increasing aggressiveness [74,75]. PDAC-derived exosomes have also been shown to possibly induce αSMA expression and manipulate pericyte phenotype, inducing vascular leak and hypoxia [76]. Only recently, have studies begun to investigate exosomes as potential biomarkers (Table 2).

### 3.1. Methods for Capturing Exosomes

Several technologies have been developed to capture and isolate circulating exosomes, each with varying degrees of sensitivity and specificity. Three primary methods, size-based, density-based, and affinity-based, have been used to isolate exosomes. For example, tunable resistive pulse sensor (TRPS) technology separates exosomes using different sized pores [80,81]. This technology also relies on differential centrifugation, a potential drawback as it may damage the exosomal membranes thus altering both the quantitative and qualitative nature of its cargo [82,83]. The utilization of centrifugation also limits this technique with regard to its widespread adaptability. To overcome this, several newer technologies have been developed. Exosomal total isolation chip (ExoTIC) is an example of an emerging platform for detecting circulation exosomes without centrifugation [84]. This technology utilizes a porous membrane that can both separate and isolate exosomes based on size. Its modular design and reproducibility have made it an attractive option for clinical application. Affinity-based methods have the ability to isolate exosomes with high purity using either antibody-coated magnetic beads targeting exosome surface proteins. Tetraspanins, a family of proteins with over 30 members (e.g., CD9, CD63, and CD81) characterized by four transmembrane domains, are commonly expressed in exosomes and are often targeted for exosome capture [85,86]. These techniques have higher specificity with regard to sorting of exomes but are associated with lower quantitative values and overall yield. Further development these techniques is ongoing and necessary.

### 3.2. Exosomes as a Screening Tool

Similar to ctDNA, exosomes have emerged as promising biomarkers for early detection of pancreatic cancer. The use of exosomes for PDAC screening has several possible advantages over ctDNA (Figure 2). First, many pancreatic cells are exocrine cells, and as such, continuously release exosomes into the blood. Second, exosomes may have a longer half-life than ctDNA. Third, given that exosomes express various surface proteins, discussed in detail below, differentiating the exosomal cell-of-origin is also possible. Several groups have compared exosome and ctDNA capture methods on samples obtained from patients with PDAC and demonstrated increased sensitivity using exosomes. For example, Allenson et al., demonstrated that in patients with localized, locally advanced, and metastatic PDAC, KRAS mutations were detected at higher percentages in peripheral blood exosomal DNA compared to ctDNA (66.7%, 80%, and 85% vs. 45.5%, 30.8%, and 57.9%, respectively) [77].

Similar to the cells from which they originate, exosomes express a wide array of proteins that can aid development of screening assays. Glypican-1 (GPC1), a cell surface proteoglycan with high expression in exosomes derived from prostate cancer cells, is one of the better studied markers that has also shown promise in PDAC. A study by Melo et al., used mass spectroscopy identify GPC1 and flow cytometry to isolate GPC1^+^ exosomes in murine pancreatic cancer models, healthy human subjects, and patients with either benign pancreatic disease as well early to late-stage PDAC. They reported near 100% sensitivity and specificity [87]. Buscail et al., performed a comprehensive study evaluating the combined diagnostic performance of CTCs and exosomes using samples obtained from patients on a prospective translational clinical trial [88]. Using both peripheral and portal blood obtained from patients with resectable PDAC, they demonstrated feasibility of capturing CD63 bead-coupled Glypican-1 (GPC1)-positive exosomes which were then combined with CRISPR/Cas9-improved KRAS quantification by ddPCR. They reported 64% of patients having GPC1^+^ exosomes in peripheral and/or portal blood. When combined, CTC and GPC1-positive-exosome detection showed 100% sensitivity, 80% of specificity, and a negative predictive value of 100% [88]. Not only was this the first study to evaluate combined CTC and exosome detection to diagnose resectable PDAC, it also demonstrated significant correlation between levels of GPC1^+^-exosomes and/or CTC presence correlated and both progression-free survival and overall survival.

Although promising, data surrounding the use of GPC1 as a biomarker for PDAC-derived exosomes is conflicting. For example, Lai et al., reported that GPC1 was not diagnostic for PDAC but they did identify 6 exosomal miRNAs that correlated with PDAC presence and were able to show subsequent decline following resection [89]. CD44v6, Tspan8, EpCAM, MET, and CD104 are other examples of cell surface proteins typically expressed in exosomes isolated from patients with PDAC and not healthy individuals [90]. Madhaven et al., demonstrated that the combination of flow cytometry coupled with RT-PCR examining microRNA expression patterns on pancreatic cancer cell exosomes could increase the screening sensitivity to 100% with a specificity of 80% [91]. Further validation in large prospective studies is needed before using exosomes for screening of PDAC can be adopted for general use.

### 3.3. Exosomes to Guide Treatment and Monitor for Recurrence

To date, few studies have explored the use of exosomes to guide treatment decisions or determine risk of recurrence in individuals who have undergone PDAC resection. Takahasi et al., isolated exosomes from patients with stage II PDAC and utilized a microarray-based miRNAs expression profiling platform to identify potential miRNA biomarkers [92]. They discovered a significant correlation between patients with elevated levels of exosomal microRNA-451a (miR-451a) and recurrence. Kawamura et al., also investigated microRNA (miR-451a, miR-4525, and miR-21) in exosomes sampled from peripheral blood and the portal vein during pancreatectomy and found that not only were levels of these miRNAs higher in portal venous blood but also that high expression was an independent prognostic factor for overall survival and disease-free survival [93]. In 2022, Bunduc et al., published a systematic review and meta-analysis on the prognostic potential of exosomes in PDAC [94]. In total, eleven studies comprising a total of 634 patients with all stages of disease were compiled. Detectable exosome miRNAs at any stage predicted increased mortality and progression and also correlated with increased mortality when identified preoperatively. The authors highlighted that the variability of study platforms likely resulted in data heterogeneity, a fundamental problem with liquid biopsy studies today. Although significant strides are being made in exosomal capture further validation and ultimately uniformization is required before these platforms are adopted into standard of care practice.

## 4. Circulating Tumor Cells

Circulating tumor cells (CTCs) are malignant cells that have undergone epithelial-mesenchymal transition and have entered the circulation either directly or through the lymphatic system [95]. These cells can be shed from primary or metastatic solid tumors and often represent a heterogenous population [96]. The half-life of these cells is between 1 to 2.4 h [97,98]. Different tumor types release varying amounts of CTCs, but the average number of cells detected in 1 mL of blood is typically fewer than 10 [99]. The primary challenge for emerging technology aimed at capturing CTCs is sorting through the millions of blood cells without causing damage to or losing CTCs. Purifying samples, so as to not contaminate with leukocytes, and identifying CTCs correctly further complicates the process of capturing these cells.

### 4.1. Methods for Capturing CTCs

Multiple technologies to capture and enrich for CTCs have recently been developed. These involve immunoaffinity methods, targeting specific antigens on the surface of tumor cells, microfluidic capture devices, and sized-based separation techniques. Immunoaffinity methods typically involve both positive enrichment for epithelial cell markers (e.g., Epithelial cellular adhesion molecule (EpCAM) or cytokeratin (CK)) and negative enrichment with CD45 to remove leukocytes. CellSearch^®^ is the only FDA-approved CTC isolation method and relies on magnetic beads coated with antibodies to EpCAM, CK, and CD45 [100]. Other similar platforms using magnetic beads include MACS^®^ and Dynabeads^®^. Tumor antigen-independent microfluidic CTC-chip technology represents another immunocapture platform that utilizes two-stage magnetophoresis and depletion antibodies against leukocytes to isolate CTCs [101]. This platform is appealing and had garnered wide-scale adoption given that pre-labeling and processing of samples prior to testing is not required. Using this technology, Nagrath et al., reported successful identification of CTCs in 115 of 116 (99%) peripheral blood samples obtained from patients with metastatic PDAC as well as lung, prostate, breast, and colon cancer [101]. They also reported a range of 5–1281 CTCs per mL and approximately 50% purity. Similar to the CTC-chip, the Herringbone-chip, passes peripheral blood through channels with micro vortices to increase the CTC exposure to EpCAM coated chip surfaces. A small study in prostate cancer described the identification of CTCs in 93% of patients with metastatic disease using this technology [102]. Lastly, several size-based separation techniques using membrane microfilters have been developed to isolate CTCs. These include isolation by size of epithelial tumor cells (ISET), ScreenCell, and ApoStream [103,104,105]. All of these methods can be used in conjunction with other DNA or RNA detection platforms. While CTCs may represent a promising biomarker assay, they likely do not fully represent the heterogenous cell population within a tumor, especially as only those cells that have undergone epithelial-mesenchymal transition will be captured.

### 4.2. CTCs as a Screening Tool

Using CTCs for early detection of PDAC remains controversial. Depending on the stage and method used, CTCs can be detected in patients with PDAC but at lower rates compared to other solid tumors [106]. Reported ranges in sensitivity are wide, while specificity typically approaches 90–100% in most studies (Table 3) [107]. Perhaps unsurprisingly, patients in whom CTCs are discovered have a worse prognosis as compared to those who do not have detectable CTCs [108]. This was highlighted in a meta-analysis by Han et al., which revealed a worse overall survival in PDAC patients with CTC-positive disease compared with those without detectable CTCs (HR = 1.23, 95% CI = 0.88–2.08, *p* < 0.001) [109]. There are also data suggesting concordance between the number of CTCs in peripheral blood and the stage of disease [110]. As one might also expect, the detection rate of CTCs is quite low in those with early-stage disease. For example, one study including patients with locally advanced PDAC reported a detection rate of only 11% [111]. Interestingly, other studies have reported that 33% to 62% of patients with premalignant lesions, such as intraductal papillary mucinous neoplasms, have detectable CTCs [112,113,114,115].

Unfortunately, comparisons to other liquid biopsy methods are limited, given that most studies have not attempted to simultaneously isolate ctDNA, exosomes, and CTCs. A large meta-analysis by Zhu et al., evaluated the diagnostic value of these three liquid biopsy methods by reviewing 19 studies involving a total of 1872 patients [122]. They reported the sensitivity, specificity and AUC for the diagnosis of PDAC using ctDNA (0.64, 0.92, and 0.94) exosomes (0.93, 0.92, and 0.98), and CTCs (0.74, 0.83, and 0.81). They argue that the lower-than-expected sensitivity for CTCs may be explained by deceased blood flow through pancreatic tissues or perhaps because CTCs are trapped in the liver. It is also possible that many tumors do not shed CTCs at early stages. Several studies have reported greater CTC yields in portal venous blood as compared to peripheral blood. For example, Tien et al., found that 58.3% of patients diagnosed with upper gastrointestinal tract cancers (68% in this study had PDAC) had CTCs detected in of portal venous blood samples compared with only 40% detected in peripheral blood [123]. These data suggest that the number of CTCs present in portal venous blood is much higher and hence can be detected more efficiently, potentially representing an alternative method for sampling.

### 4.3. CTCs to Guide Treatment and Monitor for Recurrence

CTCs may have a future role in predicting and determining treatment response. Several studies have reported that higher levels of CTCs prior to the initiation of chemotherapy predict a less robust response to treatment and reduced DFS and OS [124]. Okubo et al., conducted a prospective study including 65 PDAC patients who underwent CellSearch for isolation of CTCs and showed not only that CTC positivity was significantly greater in patients with liver metastases, but also that the presence or absence of CTCs could serve an independent prognostic factor [125]. In addition, CTC positivity 3 months after beginning therapy was 45.4% and 24.1% in those with progressive disease versus those with either stable disease or partial response, respectively. Overall survival was also significantly lower in patients with detectable CTCs after treatment (*p* = 0.045). A study by Ren et al., explored the effects of 5-fluorouracil on CTCs in an effort to identify early changes that may predict response treatment [126]. They found that Apoptotic CTCs were not only detectable but may predict response to chemotherapy. Furthermore, greater than 70% of patients with detectable CTCs prior to chemotherapy had none detected after 7 days.

The expression of cell surface proteins on CTCs has also been explored as potential predictive or prognostic biomarkers for PDAC. For example, one study evaluating 50 PDAC patients found that those with tumors expressing human mucin 1 (MUC-1) on CTCs had inferior OS compared to patients with non-MUC-1 expressing CTCs [127]. The role of cell surface proteins on CTCs has also been explored as potential marker for disease recurrence. Vimentin is an epithelial cell surface protein that has been studied by Wei et al. They demonstrated in a study of 100 patients with PDAC that increased vimentin^+^ CTCs correlated with increased disease burden in patients undergoing resection and could be used as a reliable biomarker in PDAC [121]. The utilization of CTCs as a potential risk stratification tool has been evaluated in smaller studies. The CLUSTER study, for example, prospectively measured CTCs in patients with PDAC and showed that preoperative CTC levels correlated with disease recurrence at one year in patients undergoing resection [119].

CTCs may also provide insights into mechanisms of drug resistance. Viable CTCs collected serially from patients receiving various treatments can and have been used for elucidating mechanisms of response or resistance to therapies in tissue culture. These cells can also be used for generating organoids and PDX models. For example, one study explored interactions between portal vein CTCs and immune populations and showed that CTCs could recruit immune cells and increase fibroblast differentiation [118].

Like cfDNA and exosomes, CTCs may also represent a minimally invasive method to monitor for disease recurrence. The characterization of CTCs by phenotype has been explored as a potential way of stratifying by risk of disease recurrence and OS. Poruk et al., explored aldehyde dehydrogenase (ALDH), CD133, and CD44 as markers of CTCs with a tumor-initiating cell (TIC) phenotype in patients with PDAC undergoing surgical resection [128]. The authors found that ALDH-positive CTCs and triple-positive CTCs were associated with decreased survival (*p* ≤ 0.01) and tumor recurrence. Although these data are promising, larger prospective trials are warranted to better characterize the role of CTCs in PDAC.

## 5. microRNAs

MicroRNAs (miRNAs) are highly conserved small non-coding RNAs that average 22 nucleotides in length [129]. They are naturally encoded in the genomes of various species and play important roles in the post-transcriptional regulation of their cognate mRNAs [130,131]. They modulate gene expression through repression of mRNA translation and consequently can affect cell differentiation, cell proliferation, angiogenesis, and apoptosis [132,133]. In fact, it is estimated that 2588 miRNAs regulate over 60% of the expression of human genes highlighting their importance in a diverse physiological and developmental processes as well as the pathogenesis of various human diseases including cancer [134,135,136,137]. Specifically, studies have shown that dysregulation of expression of miRNAs can lead to alteration of oncogenic and tumor suppressor pathways which in turn may contribute to cancer pathogenesis including initiation, progression, and metastasis [136,137].

Recent data indicate that miRNAs are trafficked between different subcellular compartments to regulate translation [138]. miRNAs can be secreted into circulation in extracellular fluids or shuttled to target cells via extracellular vesicles, including exosomes, microvesicles, and apoptotic bodies, enabling them to function as chemical messengers to mediate cell–cell communication [139,140,141]. They may also bind to proteins, such as the well described argonautes (AGO), such as AGO2 [139,142]. In addition to tumor cells, stem cells, macrophages, and adipocytes all have been shown to release exosomes into the circulation with specific miRNA (exomiRs) content [143,144,145,146]. Altered expression of circulating miRNAs has been shown to be associated with the origin, progression, therapeutic response, and patient survival in several tumors [147,148]. For example, the tissue specificity of miRNAs, which is required for maintaining the normal cell and tissue homeostasis, allows them to be used as potential biomarkers in diagnosing cancer of unknown primary [135,149,150,151]. Their high stability in circulation and other biofluids (e.g., cerebrospinal fluid, saliva, urine) and ease of detection have increased interest in exploiting miRNAs as novel, non-invasive biomarkers in PDAC.

### Potential Use of microRNAs in Pancreatic Cancer

Several studies have reported on the utility of circulating miRNAs as potential biomarkers in PDAC [135,147,152,153,154,155]. One study found that themiR-155-5p was not only upregulated in both tumor tissues and plasma of patients with PDAC, but levels of this miRNA were also associated with tumor stage and poor prognosis [156,157,158]. Long-term administration of gemcitabine was shown to increase overexpression of miR-155-5p, suggesting a possible resistance mechanism via anti-apoptotic activity [159]. Another miRNA, miR-373-3p, was found to be downregulated in sera obtained from patients with PDAC, and miR-373-3p level was negatively correlated with TNM stage, lymph node metastasis, and distant metastasis [160]. Furthermore, pancreatic cancer patients with reduced serum miR-373 level exhibited shorter five-year overall survival [160].

Because individual miRNAs have less discriminatory power, combining multiple miRNAs, with or without other biomarkers, has been attempted to increase specificity. The combination of miR-16 and miR-196 with CA19-9 yielded and a panel of seven miRNAs (i.e., miR-20a, miR-21, miR-24, miR-25, miR-99a, miR-185 and miR-191) has been shown to accurately differentiate PDAC from healthy sera [155]. On the other hand, a study of a three-microRNA combination, including miR-106b, miR-126 and miR-486, had a slightly reduced diagnostic accuracy [155]. Another study showed that changes in the levels of miR-25, GDF-15 and CA19-9 were highly accurate and this was further enhanced with the combination of six miRNAs, MIC-1, and CA19-9. The combination of miR-20a, miR-21, miR-25, MIC-1 and CA19-9 was even able to distinguish between pancreatic cancer and other GI/biliopancreatic diversion cancers [161]. These findings suggest that a combination of biomarkers improves diagnostic yields considerably as compared to using any single marker.

Collectively, circulating miRNAs are potentially effective non-invasive cancer biomarkers that may be used for PDAC screening, subtype classification, and drug response prediction. Nevertheless, challenges remain with respect to the sensitivity, specificity, and applicability of circulating miRNAs as biomarkers.

## 6. Challenges and Limitations of Liquid Biopsy in PDAC

Since 2016, the number of patients enrolled onto clinical trials incorporating liquid biopsies has risen dramatically (Figure 3). Despite promising advances, several limitations continue to hamper the incorporation of liquid biopsy into standard clinical practice for patients with PDAC. Assay sensitivities remain variable and, depending on modality, are impacted by clinical stage, tumor burden, location of primary and metastatic sites, and rates of cell turnover [55,56,57,63]. Sensitivity of modern platforms such as CAPPSeq, iDES and SAFE-seq have risen considerable and now able to detect VAFs < 0.01% [162]. Notwithstanding, the rates of false negatives is high, especially in patients with localized disease [61,63]. In 2021, two PDAC ctDNA screening platforms became commercially available, and although neither are FDA approved, with further validation these may soon provide clinicians additional tools for screening high risk patients [163].

Although liquid biopsies are becoming commonplace in cancer care, for example to rapidly detect potentially actionable mutations (e.g., adenocarcinoma of the lung) or to detect recurrence (e.g., resected colon cancer), their role in PDAC is less defined. Due to the limited number of actionable somatic mutations in PDAC, germline sequencing remains more impactful than somatic sequencing. In addition, CA19-9 is a readily available, inexpensive, and validated tool to approximate of disease burden and trend response/screen for recurrence. This makes it difficult to justify the higher costs associated with serial liquid biopsy. CA19-9 detection also has the benefit of being somewhat comparable across testing sites, whereas liquid biopsy techniques currently vary significantly based on cellular component tested, method of genetic sequencing, array of genes examined, and platform used. There are also concerns that ctDNA may not capture true tumor genetic heterogeneity. Select tumor clones may release more or less amounts of ctDNA based on their location and cell turnover rate. For example, lung and brain metastases may produce less ctDNA than other metastatic sites [165,166].

There are a growing number of studies showing that liquid biopsy can accurately predict tumor response and that sensitivity and specificity are increased when combined with CA19-9 [18,50,58,62,64]. However, there is limited data indicating that incorporation of liquid biopsy into clinical practice improves clinical outcomes or reduces unnecessary chemotherapy administration. Even in resected colorectal cancer, ctDNA analysis using Signatera^TM^ has yet to clearly demonstrate survival benefit as a surveillance strategy when compared to standard of care imaging [166]. Currently, there are multiple ongoing prospective trials assessing the clinical utility of liquid biopsies in PDAC, but until these are completed and the data is fully analyzed, routine use of liquid biopsy for diagnosis, monitoring of treatment response, and detection of recurrence PDAC is unsupported (Table 4).

## 7. Conclusions

Over the past few years, significant and exciting progress been made in liquid biopsy technology. These non-invasive platforms hold considerable promise and are likely to soon improve PDAC diagnosis, treatment monitoring, prognostication, and surveillance. Furthermore, liquid biopsies may provide key insights into tumor evolution and mechanisms of drug resistance. Despite these advances, however, adoption in routine clinical practice has been thwarted by insufficient sensitivity and specificity, especially in early-stage disease. Comprehensive validation and standardization of contemporary liquid biopsy methodologies as well as confirmation in large prospective randomized controlled trials are necessary and ongoing. Given the enormous potential to change clinical practice, liquid biopsy technology is rapidly improving and will soon become a future standard tool in PDAC management.

## Figures and Tables

**Figure 1 ijms-24-04238-f001:**
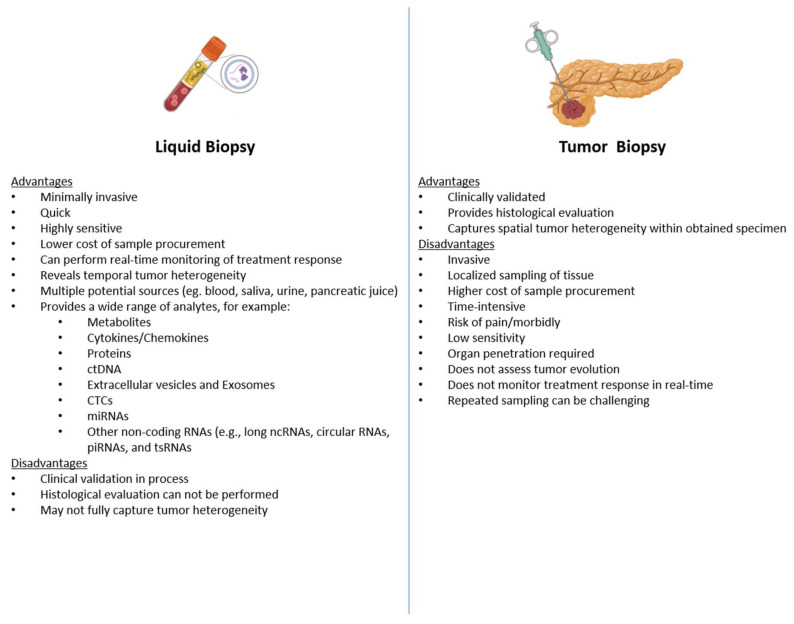
Liquid versus Tumor Biopsy. Abbreviations: DNA, deoxyribonucleic acid; PCR, polymerase chain reaction; ctDNA, circulating tumor DNA; CNA, copy number alterations; NGS, next generation sequencing; ddPCR, droplet digital PCR; RNA, ribonucleic acid; CTC, circulating tumor cells.

**Figure 2 ijms-24-04238-f002:**
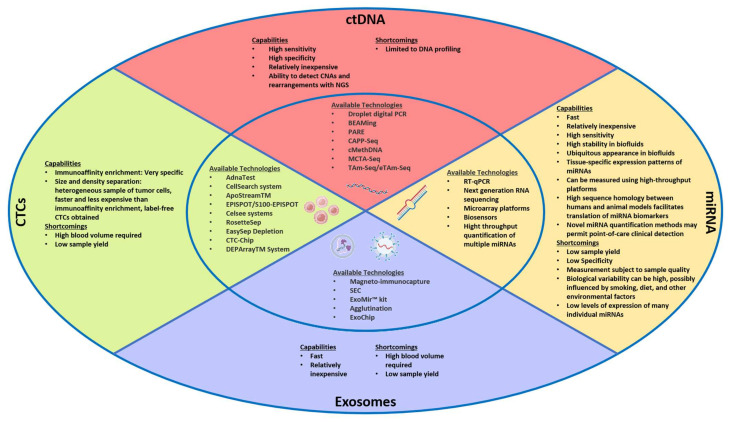
Comparison of liquid biopsy techniques. Abbreviations: DNA, deoxyribonucleic acid; PCR, polymerase chain reaction; ctDNA, circulating tumor DNA; CNA, copy number alterations; NGS, next generation sequencing; ddPCR, droplet digital PCR; RNA, ribonucleic acid; CTC, circulating tumor cells; miRNA, microRNA.

**Figure 3 ijms-24-04238-f003:**
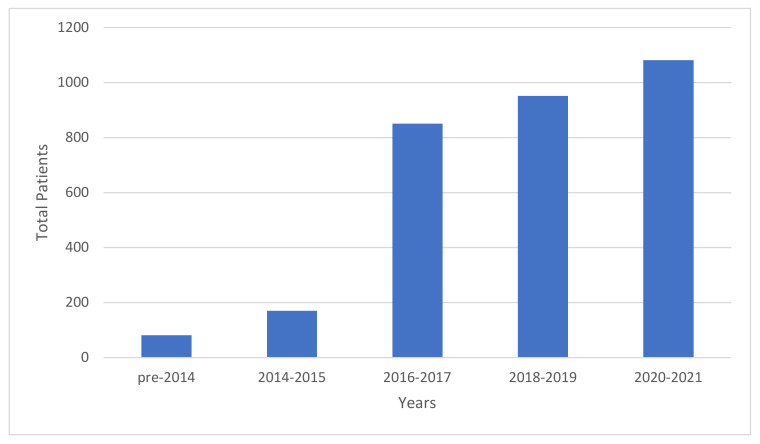
Patients enrolled onto clinical trials assessing the role of liquid biopsy in PDAC. Grouped per 2-year increments [13,17,18,26,36,48,49,50,51,52,53,54,55,56,57,58,59,60,61,62,63,65,66,67,68,69,77,78,79,100,101,116,117,164].

**Table 1 ijms-24-04238-t001:** Studies of ctDNA-based liquid biopsy in pancreatic ductal adenocarcinoma.

Study	No. of pts	Stage	Platform	Markers	Findings
Sorenson et al., 1994 [48]	3	Stage IV PDAC	PCR and allele specific amplification	*KRAS*	*KRAS* mutation detected in 3/3 PDAC pts and 0/5 pts without PDAC.
Maire et al., 2002 [49]	47	PDAC, all stages	qPCR and allele specific amplification	*KRAS*	*KRAS* mutation detected in 47% of 47 PDAC pts and 13% of 31 controls with chronic pancreatitis (*p* < 0.002). The combination of KRAS and CA19-9 gave a sensitivity and specificity of 98% and 77%, respectively for identifying PDAC pts.
Zill et al., 2015 [26]	18	Advanced pancreato-biliary cancers	NGS	*KRAS*, *TP53*, *APC*, *SMAD4*, and *FBXW7*	Mutations detected in 90.3% of pts.
Sausen et al., 2015 [50]	77	Stage II PDAC	NGS and ddPCR	*KRAS*	*KRAS* mutation detected in 43% of pts. Detection of ctDNA after resection predicted clinical relapse and poor prognosis. ctDNA could detect recurrence 6.5 months earlier than CT.
Kinugasa et al., 2015 [51]	75	PDAC, all stages	ddPCR	*KRAS*	*KRAS* mutation rate 74.7% in tissue and 62.6% in ctDNA from 75 pts with PDAC. OS significantly longer in pts without detectable *KRAS* mutations in ctDNA.
Berger et al., 2016 [52]	24	Metastatic PDAC	ddPCR	*KRAS*	*KRAS* mutation detected in 41.7% of 24 PDAC pts and 0/101 pts with IPMN, borderline IPMN, resected SCAs or no pathology.
Henriksen et al., 2016 [53]	95	PDAC, all stages	Methylation-specific PCR	*BMP3*, *RASSF1A*, *BNC1*, *MESTv2*, *TFPI2*, *APC*, *SFRP1* and *SFRP2*	Mean methylated genes detected in 95 PDAC pts was 8.41 compared with 4.74 in control group of 97 pts with chronic pancreatitis, 59 with acute pancreatitis and 27 without evidence of PDAC (*p* < 0.001). A diagnostic prediction model using age and methylation pattern had 76% sensitivity 76% and 83% specificity for detecting PDAC.
Tjensvoll et al., 2016 [54]	14	PDAC, all stages	PNA-clamp PCR	*KRAS*	*KRAS* mutation detected in 71% of pts with PDAC. Pre-therapy ctDNA predicted decreased DFS and OS. Changes in ctDNA levels correlated with radiological findings and CA19-9 levels.
Cheng et al., 2017 [55]	188	Metastatic PDAC	NGS and ddPCR	*KRAS*, *BRCA2*, *EGFR* and *KDR*	*KRAS* mutation detected in 72% of pts.
Pietrasz et al., 2017 [56]	135	PDAC, all stages	NGS	*KRAS*	ctDNA was detected in 48% of pts with advanced PDAC, and was an independent prognostic marker in advanced PDAC. ctDNA was associated with shorter DFS and OS when detected after resection of localized PDAC.
Cohen et al., 2017 [57]	221	Resectable PDAC	ddPCR	*KRAS*	*KRAS* mutation detected in 30% of 221 pts with resectable PDAC and 1/182 control pts. All *KRAS* mutations in ctDNA concordant with those detected in tissue. Screening for PDAC using ctDNA positivity or elevation of any of 4 protein biomarkers cutoffs produced a sensitivity of 64% and specificity of 99.5%.
Sugimori et al., 2019 [58]	45	Metastatic PDAC	dPCR	*KRAS*	In the 6 locally advanced cases, KRAS mutation appeared concurrently with liver metastasis. Among the 6 cases with liver metastasis, KRAS mutation disappeared during the duration of stable disease or a partial response, and reappeared at the time of progressive disease.
Patel et al., 2019 [59]	112	Metastatic PDAC	NGS	*KRAS, TP53*	Concordance for KRAS alterations between ctDNA and tissue DNA from metastatic sites was significantly higher than between ctDNA and primary tumor DNA (72% vs. 39%, *p* = 0.01). Higher VAF was an independent prognostic factor for worse OS (HR, 4.35; 95% confidence interval: 1.85–10.24).
Lee et al., 2019 [60]	112	Resectable PDAC	ddPCR	*KRAS*	Patients with ctDNA-positive status postoperatively had reduced DFS and (HR 5.4; *p* < 0.0001) and OS (HR 4.0; *p* = 0.003).
Guler et al., 2020 [36]	64	PDAC, all stages	Methylation-specific NGS	Many	Differential hydroxymethylation of genes related to pancreas development or function and cancer pathogenesis were identified in a discovery cohort and used to differentiate PDAC-associated from normal plasma samples in a validation cohort.
Jiang et al., 2020 [61]	27	Resectable PDAC	NGS	*KRAS*, *TP53*	Patients with ctDNA-positive status postoperatively had reduced DFS compared to those with ctDNA-negative status (HR, 5.20; *p* = 0.019).
Toledano-Fonseca et al., 2020 [62]	61	Metastatic PDAC	dPCR	*KRAS*	Higher RAS VAF and higher cfDNA levels correlated with worse DFS and OS. The combination of CA19-9 with VAF, cfDNA levels improved prognostic stratification.
Uesato et al., 2020 [63]	104	Metastatic PDAC	NGS	Many	Detectable ctDNA correlated with worse PFS and OS. ctDNA correlated with number of liver metastases and with the presence of metastases at other sites.
Wei et al., 2020 [64]	70	Metastatic PDAC	NGS	Many	Higher VAF was associated with increased CA19-9, metastasis, and worse prognosis. CNAs were concordant over time even in patients with progressive disease. Patients with more abundant baseline CNAs exhibited a better response to chemotherapy.
Nakamura et al., 2020 [65]	363	Metastatic PDAC	NGS	*TP53*, *KRAS* and *GNAS*	PDAC patients had lower MSI prevalence and TMB score, but higher rates of germline *BRCA* mutations than patients with gastroesophageal and colon cancer.
Bachet et al., 2020 [66]	113	Metastatic PDAC	NGS	*TP53*, *KRAS* and others	ctDNA was positive at baseline in 68% of patients. Detectable ctDNA was an independent negative prognostic factor for OS and PFS. Early change in ctDNA levels correlated with ORR.
Hussung et al., 2021 [67]	25	Resectable PDAC	ddPCR	*KRAS*	Integration of cell free mutant *KRAS* cfDNA levels and CA19–9 levels outperformed either individual marker when predicting PFS and OS.
Pietrasz et al., 2022 [68]	255	Metastatic PDAC	ddPCR	*HOXD8* and *POU4F1* methylated markers	56.8% of patients were ctDNA positive. Median PFS and OS were 5.3 and 8.2 months in ctDNA-positive and 6.2 and 12.6 months in ctDNA-negative patients, respectively. ctDNA positivity was associated with young age and high CA19-9 level and was an independent prognostic marker for PFS and OS.
Renouf et al., 2022 [69]	174	Metastatic PDAC	NGS	*KRAS* and others	*KRAS* wild type metastatic PDAC may derive benefit from immunotherapy.
Huang et al., 2022 [18]	74	Stage III-IV PDAC	NGS	*KRAS*	cfDNA concentration of >9.71 ng/mL before and after first two courses of chemotherapy was strongly predictive of the development of new distant metastasis (NDM) on CT scans 3 months later (accuracy 94.4%, AUC 0.971, *p* < 0.0001).

CA19-9, carbohydrate antigen 19-9; CAN, copy number alteration; HR, hazard ratio; DFS = disease free survival; OS, overall survival; PFS, progression free survival; VAF, variant allele frequency; NGS, next generation sequencing; PCR, polymerase chain reaction; ddPCR, droplet digital PCR; qPCR, quantitative PCR; CTCs, circulating tumor cells.

**Table 2 ijms-24-04238-t002:** Studies of exosomal liquid biopsy in pancreatic ductal adenocarcinoma.

Study	No. of pts	Type of pts	Isolated Component	Platform	Markers	Findings
Allenson et al., 2017 [77]	68	PDAC, all stages	ctDNA and exomes	ddPCR	*KRAS*	*KRAS* mutation in exoDNA, identified in 7.4%, 66.7%, 80%, and 85% of age-matched controls (54), localized, locally advanced, and metastatic PDAC pts, respectively. *KRAS* mutation in cfDNA was detected in 14.8%, 45.5%, 30.8%, and 57.9% of these individuals. Higher exoKRAS VAFs associated with decreased DFS in pts with localized disease.
Yang et al., 2017 [78]	48	Resectable PDAC	Exomes	dPCR	*KRAS*, *TP53*	*KRAS* and *TP53* mutations identified in exosomal DNA of 39.6% and 4.2% of PDAC cases, respectively. In 114 healthy controls 2.6% and 0% had *KRAS* and *TP53* mutations in exosomal DNA, respectively.
Bernard et al., 2019 [17]	194	PDAC, all stages	ctDNA and exomes	ddPCR and NGS	*KRAS*	In 34 pts with potentially resectable PDAC, increase in exoDNA level after neoadjuvant therapy was significantly associated with disease progression (*p* = 0.003). Concordance rate of *KRAS* mutation between liquid biopsy and surgical resection >95%. In stage IV PDAC, detectable ctDNA correlated with decreased DFS and OS.
Castillo et al., 2018 [79]	103	PDAC, all stages	Exomes	ddPCR	*KRAS*, exosome surface proteins	*KRAS* mutation in exoDNA of 73% of PDAC pts following exosome capture using selected biomarkers.

VAF, variant allele frequency; NGS, next generation sequencing; PCR, polymerase chain reaction; ddPCR, droplet digital PCR; qPCR, quantitative PCR.

**Table 3 ijms-24-04238-t003:** Studies of circulating-tumor cells in pancreatic ductal adenocarcinoma.

Study	No. of pts	Type of pts	Platform	Markers	Findings
Allard et al., 2004 [100]	16	Stage IV PDAC (within study of multiple cancer types)	Antibody detection	CellSearch system	19% of 21 samples from 19 pts with stage IV PDAC had ≥2 CTCs per 7.5 mL blood. In the larger cohort of patients with metastatic carcinoma from various sites, 36% (781/2183) pts had had ≥2 CTCs per 7.5 mL blood
Nagrath et al., 2007 [101]	15	Stage IV PDAC (within study of multiple cancer types)	Antibody detection	CTC-chip system using EpCAM-coated microposts	CTCs identified in 115/116 pts with metastatic carcinoma (15/15 PDAC pts). Temporal changes in CTC numbers correlated with radiologic response/progression.
Kulemann et al., 2017 [116]	58	PDAC, all stages	CTC detection based on size and genomic analysis	Size-based CTC isolation	CTCs identified in 53/58 pts with PDAC and 0/10 healthy controls. Pts with >3 CTCs/mL had non-significantly reduced OS. KRAS mutations in CTCs were discordant with those in primary tumor in 11/26 pts and concordant in 15/26 pts with KRAS mutated CTCs. Pts with KRAS mutations in CTCs had longer OS than other pts.
Effenberger et al., 2018 [117]	69	PDAC, all stages	Antibody detection	Anti-cyto-keratin/anti-EpCAM staining	CTCs identified in 23/69 pts with PDAC, ranging from 1–19 cells. Detectable CTCs correlated with decreased DFS and OS.
Arnoletti et al., 2018 [118]	11	Resectable PDAC	Antibody detection	FACS-isolation system	CTC proliferation and resistance to T cell cytotoxicity were decreased among patients who received neoadjuvant chemotherapy.
Gemene-tzis et al., 2018 [119]	200	PDAC, all stages	Size followed by antibody detection	ISET assay	Neoadjuvant chemotherapy and surgery both significantly lowered total CTCs.
Liu et al., 2018 [120]	29	Advanced PDAC	Antibody detection	EasySep Human CD45 Depletion Kit	Absolute number of CTCs in portal vein was significantly higher than that in peripheral circulation and was associated with intrahepatic metastases and poor prognosis.
Wei et al., 2019 [121]	76	PDAC, all stages	Density followed by antibody detection	CytoQuest CR system	Combined vimentin+ CTCs and CA19-9 identified PDAC cases with an area under the curve of 0.968.
Chen et al., 2022 [115]	80	PDAC and IPMNs	Antibody detection	NE-imFISH	CTCs isolated from 80 patients with increased specificity/sensitivity in detecting PDAC compared to CA19-9 alone.
Dopico et al., 2022 [114]	8	Resectable PDAC	ctDNA by ddPCR, CTCs by antibody detection	Microfluidic droplet digital PCR (DDPRC)	Both cfDNA and CTCs (81% and 91%, respectively) can be isolated in patients after starting neoadjuvant therapy and before surgical resection.

CA19-9, carbohydrate antigen 19-9; DFS = disease free survival; OS, overall survival; PCR, polymerase chain reaction; ddPCR, droplet digital PCR; qPCR, quantitative PCR; CTCs, circulating tumor cells.

**Table 4 ijms-24-04238-t004:** Ongoing prospective clinical trials with primary outcomes evaluating the utility of liquid biopsy.

Date Initiated	Analyte	No. of pts	Disease Setting	Trial Number
12 June 2019	CTCs	50	Resectable PDAC	NCT04289961
7 November 2017	ctDNA, Exosomes, CTCs	700	Screening/and response to treatment	NCT03334708
15 August 2017	ctDNA, Exosomes	100	PDAC screening	NCT0325½8
2 January 2019	Exosomes	200	PDAC screening	NCT03791073
15 March2015	Exosomes	111	PDAC (any stage)	NCT02393703
3 November 2022	ctDNA	150	PDAC (any stage)	NCT05604573
18 February 2019	ctDNA	40	Resectable PDAC	NCT03435536
29 January 2020	ctDNA	200	Resectable PDAC	NCT04246203
1 June 2022	ctDNA	200	Resectable PDAC	NCT05400681
21 September 2021	ctDNA	15	PDAC (any stage)	NCT05497531
29 July 2022	ctDNA	150	Resectable PDAC	NCT05479708
15 June 2022	ctDNA	50	Resectable PDAC	NCT05052671

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
