# Peer review of "Advances in Liquid Biopsy Technology and Implications for Pancreatic Cancer"

_ijms, 2023, doi:10.3390/ijms24044238_

Round 1

Reviewer 1 Report

Advances in Liquid Biopsy Technology and Implications for Pancreatic Cancer is a literature review whose objectives are to present the current state of knowledge on the analysis of liquid biopsies concerning pancreas cancer.

Pancreatic adenocarcinoma with its limited molecular landscape is an ideal candidate for mutation research in liquid biopsy in order to be used for population screening, determination of treatment response or resistance, and detect recurrence. These advances would allow to personalize and adapt patients’ treatment.

This article presents three liquid biopsy techniques (ctDNA, Exosomes, CTCs) with the characteristics, advantages and defects of each. It describes also the methods for detecting and analyzing them, their abilities as a screening tool, guiding treatment and monitoring recurrence by reviewing nineteen articles. Finally, it highlights the lack of sensitivity and specificity of mutation research for pancreatic cancer in clinical practice despite promising leads.

The article is well written and well organized to directly understand its aim. The only figure of the article is very clear to compare liquid biopsy techniques.

The different sections are well developed and the authors made a good synthesizing of the literature in Table 1. This review provides a good representation of the state of knowledge on liquid biopsy technology for pancreatic adenocarcinoma with recent and relevant cited references.

However, two recent and similar reviews were already published: “Early detection of pancreatic cancer: Where are we now and where are we going?” Zhou et al 2017 and “Early screening and diagnosis strategies of pancreatic cancer: a comprehensive review” Yang et al 2021 are two Chinese studies who already have shown the same conclusions on the lack of sensitivity and specificity with current techniques of liquid biopsy with different articles cited. These two articles are not cited in this paper.

The part of microRNA in this article is not well described despite the interest it arouses in literature and should be presented more clearly than it is in the paragraph between line 285 and 302.

Author Response

We appreciate the extensive comments and suggestions provided by the reviewer. In response, we have heavily edited and extensively revised the prior draft of the manuscript. The reviewer recommended adding several recently published Chinese studies to the manuscript which was done. In addition, we have added an additional 9 publications from 2022 and 11 publications from 2021 on this topic to increase relevance and provide a more up-to-date discussion. In addition, a fourth section on microRNA was added, as was suggested by this reviewer.

Reviewer 2 Report

The manuscript by Alexander G Raufi et al describes “Advances in Liquid Biopsy Technology and Implications for Pancreatic Cancer”. In my opinion, the manuscript is well-written text with a logical introduction and a wide literature study. However, my impression is, that manuscript raises a similar topic to many other articles (listed below), and I’m not convinced of the unique approach to the subject.

Similar reviews

Schlick K, Kiem D, Greil R: Recent Advances in Pancreatic Cancer: Novel Prognostic Biomarkers and  Targeted Therapy-A Review of the Literature. Biomolecules 11, 2021

Luo G, Jin K, Deng S, et al: Roles of CA19-9 in pancreatic cancer: Biomarker, predictor and promoter.  Biochim Biophys Acta Rev Cancer 1875:188409, 2021

Pietri E, Balsano R, Coriano M, et al: The implication of liquid biopsies to predict chemoresistance in pancreatic cancer. Cancer Drug Resist 4:559-572, 2021

Heredia-Soto V, Rodriguez-Salas N, Feliu J: Liquid Biopsy in Pancreatic Cancer: Are We Ready to Apply It in the Clinical Practice? Cancers (Basel) 13, 2021

Xu W, Wen Y, Liang Y, et al: A plate-based single-cell ATAC-seq workflow for fast and robust profiling of chromatin accessibility. Nat Protoc 16:4084-4107, 2021

Pteri et al. 2020 Circulating Cell-Free DNA-Based Liquid Biopsy Markers for the Non-Invasive Prognosis and Monitoring of Metastatic Pancreatic Cancer

Kaczor-Urbanowicz KE, Cheng J, King JC, et al: Reviews on Current Liquid Biopsy for Detection and Management of Pancreatic Cancers. Pancreas 49:1141-1152, 2020

Zhu Y, Zhang H, Chen N, et al: Diagnostic value of various liquid biopsy methods for pancreatic cancer: A systematic review and meta-analysis. Medicine (Baltimore) 99:e18581, 2020

Moreover, the title is Advances in Liquid Biopsy Technology and Implications for Pancreatic Cancer, whereas from 2020/2021/2022 I have found only three original articles in the author's reference however two of them were previously mentioned by other reviews authors:

Toledano-Fonseca M, Cano MT, Inga E, et al: Circulating Cell-Free DNA-Based Liquid Biopsy Markers for 500 the Non-Invasive Prognosis and Monitoring of Metastatic Pancreatic Cancer. Cancers (Basel) 12, 2020

Swanson E, Lord C, Reading J, et al: Simultaneous trimodal single-cell measurement of transcripts, epitopes, and chromatin accessibility using TEA-seq. Elife 10, 2021

Sugimori M, Sugimori K, Tsuchiya H, et al: Quantitative monitoring of circulating tumor DNA in patients 498 with advanced pancreatic cancer undergoing chemotherapy. Cancer Sci 111:266-278, 2020

Minor problems are "articles", "commas", and in line 393: "...represent a minimal invasive method", should be “minimally?

My opinion does not judge the general quality of the manuscript or any signs of plagiarism but rather the repetition of articles on similar issues. I recommend considering the manuscript after major changes, where the authors should prepare and submit an updated version of their manuscript.

Author Response

We appreciate the extensive comments and suggestions provided by the reviewer. As techniques for liquid biopsy improve and applications broaden, it is clear that there is increasing interest in evaluating a potential role pancreatic cancer. As the reviewer points out, there have a been several recent reviews on similar topics.  We have extensively revised and updated the manuscript, addressing grammatical and semantic errors, as well as adding a new section on microRNAs. In addition, we have updated the contents to include 9 publications from 2022 and 11 publications from 2021. Lastly, we have also added a section dedicated to the challenges and limitations of liquid biopsy, specifically in pancreatic cancer. 

Reviewer 3 Report

Dear authors.

Thanks for your working for performing so complicated reviews about the liquid biopsy in pancreatic cancers. 

However, according to our knowledge, the same review of lipid biopsy in pancreatic cancers just be published several ago.

I am gland to read the newer documents from your research team in the future.

Best regards.     

Author Response

We thank the reviewer for the time and effort put into reviewing the article. We have updated it significantly, with several new sections, and 20 additional articles from 2021 and 2022 were added. In 2020, IJMS did print a review on ctDNA in pancreatic cancer titled “Current Status of Circulating Tumor DNA Liquid Biopsy in Pancreatic Cancer” however, this review did not cover exosomes, circulating tumor cells, microRNAs, areas in which strides have recently been made. These topics are reviewed here-in, in detail, and we have included a section directly addressing some of the challenges and limitations of such approaches.

Reviewer 4 Report

This manuscript addresses the new goals of liquid biopsy in the treatment of pancreatic cancer. The work is really well structured, making excursus on ctDNAs, exosomes and CTCs, as well as the techniques currently used for the analysis of these substrates. For each method and biological substrate treated, a careful correlation is made in the use for the diagnosis of pancreatic cancer. The review is well written and structured and well summarizes the information already existing in the literature. I believe it gives added value to the field of precision medicine and therefore my review is fully positive suggesting the acceptance of this work without the need to make any changes.

Author Response

We appreciate the reviewer's comments and here-in submit a revised manuscript with significant additions which will further enhance the articles quality and breadth of coverage.

Round 2

Reviewer 2 Report

Despite many changes still, I am still not convinced and my opinion is that duplicating articles covering exactly the same aspects in so short period of time limits the contribution of the article to the field. I support my view that  paper is practical, comprehensive and well-written, however, the authors should add/develop some new figures, points i.e. clinical trials stage, or perhaps a new section with an analysis of problems should be developed. Below list of the articles (some in manuscript reference) covering all aspects mentioned by the authors. Moreover, I'm wondering if the miRNA section was developed so perhaps figure 1 should be updated?

Schlick K, Kiem D, Greil R: Recent Advances in Pancreatic Cancer: Novel Prognostic Biomarkers and  Targeted Therapy-A Review of the Literature. Biomolecules 11, 2021

Includes miRNA

Pietri E, Balsano R, Coriano M, et al: The implication of liquid biopsies to predict chemoresistance in pancreatic cancer. Cancer Drug Resist 4:559-572, 2021

Includes CTC, EXOSOMES, CTDNA

Heredia-Soto V, Rodriguez-Salas N, Feliu J: Liquid Biopsy in Pancreatic Cancer: Are We Ready to Apply It in the Clinical Practice? Cancers (Basel) 13, 2021

Include also miRNA

Kaczor-Urbanowicz KE, Cheng J, King JC, et al: Reviews on Current Liquid Biopsy for Detection and Management of Pancreatic Cancers. Pancreas 49:1141-1152, 2020

Include CT, CTC and miRNA

Author Response

We again appreciate the reviewer's comments and once again have expanded the article. Although one cannot argue against the fact that there were ~15 reviews published on liquid biopsies in PDAC between 2021 and 2022 (compared to the over 50 reviews on liquid biopsy in breast cancer in 2022 alone), we believe this review to be extensive, not only incorporating a comprehensive review on ctDNA, exosomes, miRNA, and CTCs, but also incorporating all of the major studies from 2022. A review on this topic has not been published in the last 6 months. As the reviewer suggested, in an effort to further develop the paper we added two new figures, one comparing liquid biopsy to tissue biopsy and another comparing the four aforementioned liquid biopsy techniques. We have updated the previously labeled figure 1, now figure 2, to include miRNAs. The existing tables have been updated to include newer studies that were published in 2022. A new section focused on the challenges and limitations of liquid biopsy in PDAC was also added. We have also included an additional table on current ongoing trials and finally, we added a graph which depicts the number patients with PDAC onto clinical trials evaluating liquid biopsies over the past decade. We hope that these additions will make this paper worthy of publication in 2023.

Reviewer 3 Report

Dear authors:

  Thanks for your working hard for revising this article.

  After this procedure, this document become easily realizing for the readers.

Best regards 

Author Response

We again appreciate the reviewer's comments and once again have expanded the article. In an effort to further develop the paper we added two new figures, one comparing liquid biopsy to tissue biopsy and another comparing the four aforementioned liquid biopsy techniques. We have updated the previously labeled figure 1, now figure 2, to include miRNAs. The existing tables have been updated to include newer studies that were published in 2022. A new section focused on the challenges and limitations of liquid biopsy in PDAC was also added. We have also included an additional table on current ongoing trials and finally, we added a graph which depicts the number patients with PDAC onto clinical trials evaluating liquid biopsies over the past decade. 

Round 3

Reviewer 2 Report

I appreciate the authors' efforts. The paper is practical, comprehensive, and well-written however to the editorial decision I'm leaving the final decision if the article has a significant contribution to the field.